# Melamine Foam-Derived Carbon Scaffold for Dendrite-Free and Stable Zinc Metal Anode

**DOI:** 10.3390/molecules28041742

**Published:** 2023-02-11

**Authors:** Yong Liu, Feng Tao, Yibo Xing, Yifei Pei, Fengzhang Ren

**Affiliations:** 1School of Materials Science and Engineering, Provincial and Ministerial Co-Construction of Collaborative Innovation Center for Non-Ferrous Metal New Materials and Advanced Processing Technology, Henan University of Science and Technology, Luoyang 471023, China; 2Science & Technology Innovation Center for Advanced Materials of Intelligent Equipment, Longmen Laboratory, Luoyang 471023, China; 3Henan Key Laboratory of Non-Ferrous Materials Science & Processing Technology, School of Materials Science and Engineering, Henan University of Science and Technology, Luoyang 471023, China

**Keywords:** melamine foam, carbon foam, Zn metal anode, Zn dendrite, electrochemical performance

## Abstract

Aqueous Zn-ion batteries (AZIBs) are one of the most promising large-scale energy storage devices due to the excellent characteristics of zinc metal anode, including high theoretical capacity, high safety and low cost. Nevertheless, the large-scale applications of AZIBs are mainly limited by uncontrollable Zn deposition and notorious Zn dendritic growth, resulting in low plating/stripping coulombic efficiency and unsatisfactory cyclic stability. To address these issues, herein, a carbon foam (CF) was fabricated via melamine-foam carbonization as a scaffold for a dendrite-free and stable Zn anode. Results showed that the abundant zincophilicity functional groups and conductive three-dimensional network of this carbon foam could effectively regulate Zn deposition and alleviate the Zn anode’s volume expansion during cycling. Consequently, the symmetric cell with CF@Zn electrode exhibited lower voltage hysteresis (32.4 mV) and longer cycling performance (750 h) than the pure Zn symmetric cell at 1 mA cm^−2^ and 1 mAh cm^−2^. Furthermore, the full battery coupling CF@Zn anode with MnO_2_ cathode can exhibit a higher initial capacity and better cyclic performance than the one with the bare Zn anode. This work brings a new idea for the design of three-dimensional (3D) current collectors for stable zinc metal anode toward high-performance AZIBs.

## 1. Introduction

With the rapid developments of electric vehicles and smart grids, the ever-increasing demand for lithium-ion batteries (LIBs) and their further application would be hindered by the limited lithium resource. In this context, aqueous Zn-ion batteries (AZIBs) have been developed as alternatives to LIBs because of their low-cost, non-toxic, and high safety [1,2,3,4]. Nevertheless, during the charge-discharge process, the Zn metal anodes suffer from uncontrolled dendritic growth and harmful side reactions, leading to the unsatisfactory electrochemical performance of AZIBs [5,6]. In particular, the uncontrolled Zn dendritic growth may puncture the separator of the cell and cause an internal short circuit, which results in low Coulombic efficiency, poor reversibility and rapid capacity degradation of AZIBs, which severely limits their further application and development [7,8].

Until now, many efforts have been devoted to solving the aforementioned problems, including surface modification of Zn metal anodes [9,10,11,12,13,14,15,16], electrolyte optimization [17,18,19], novel separator design [20] and three-dimensional (3D) host design [21,22,23,24]. For example, He et al. [25] deposited an ultrathin Al_2_O_3_ film on the Zn anode via the atomic layer deposition method, which effectively improved the interfacial wettability of the Zn anode and thus physically suppressed the formation of Zn dendrites. However, the presence of this insulating layer increases the internal impedance of the cell, hinders electron transport, and reduces the kinetics of the cell redox reaction [26]. To tackle this problem, some highly conductive carbon-based materials have been coated or composited with Zn anodes, such as hydrogen-substituted graphdiyne [14], zinc microspheres/carbon nanotubes/nanocellulose composite film [15], and carbon-coated NaTi_2_(PO_4_)_3_ [16], successfully regulating uniform Zn deposition. For example, a dual-functional carbon-coated NaTi_2_(PO_4_)_3_ (NTP-C) artificial protective layer with a large surface area was constructed onto the surface of metallic Zn by Wu and co-workers [16]. Benefiting from a synergistic strategy, NTP-C coating not only takes advantage of carbon to provide abundant Zn deposition sites to homogenize nucleation, adjust electric field distribution, and reduce local current density but also utilizes the ionic channel in NTP structure to modulate the distribution of Zn^2+^ flux at the same time. As a result, the NTP-C@Zn//α-MnO_2_ full cell exhibited enhanced electrochemical performance for 1200 cycles with a capacity retention of 76.6% under 5 C [16]. In addition, electrolyte additive strategies have been widely used to inhibit zinc dendrite growth in recent years. For instance, Wang et al. [18] used high concentrations of Zn(TFSI)_2_ and LiTFSI as electrolytes for aqueous Zn-ion batteries, which effectively promoted dendrite-free deposition and stripping of Zn ions. However, most of the highly concentrated electrolyte additives are expensive, toxic and difficult to prepare, which is not applicable for practical application [27]. Compared with the abovementioned two strategies, the 3D host design is considered a simple and effective approach to induce uniform Zn ion deposition. The used 3D current collectors have a large specific surface area, which allows sufficient contact with the electrolyte and uniform electric field distribution, thus ensuring uniform zinc ion distribution and alleviating zinc dendrite generation [28,29,30]. Recently, Lin and his co-workers [31] fabricated a freestanding 3D Zn-graphene anode to mitigate Zn dendrite growth. Unfortunately, the high price (~¥ 132 g^−1^, price from Suzhou Tanfeng Graphene Technology Co., Ltd.) and complicated preparation process of this 3D Zn-graphene anode limit its large-scale application. As a type of 3D carbon host, carbon foam (CF) possesses a 3D interconnected structure and could facilitate the transport of ions, which has been used for various electrochemical energy storage applications [16]. Melamine foam, a commercially available material, has been widely used in the chemical industry due to its low cost (~¥ 2.6 g^−1^, price from SINOYQX Co., Ltd., Chengdu, China), low density, high porosity, and open structures [32]. The melamine foam-derived CF has rich N-containing functional groups, such as pyridine nitrogen, pyrrole nitrogen and C=O [33], which have been proven to be highly zincophilic [34]. Moreover, these N-containing functional groups on CF would effectively control uniform Zn deposition. To the best of our knowledge, the melamine foam (MF) derived CF has been rarely reported as a 3D host for Zn metal anode, and especially its Zn deposition behavior has not been systematically investigated.

Herein, we fabricated a carbon foam (CF) by simple calcination of inexpensive melamine foam and investigated it as a 3D scaffold for zinc anode. The abundant functional groups on the surface of CF could effectively promote the Zn uniform deposition and alleviate the dendritic growth. Therefore, the CF@Zn symmetric cell exhibited lower overpotential (32.4 mV) and long cycle life (750 h) than the pure Zn symmetric cell at a current density of 1 mA cm^−2^ and a capacity of 1 mAh cm^−2^. Furthermore, the full cell with CF@Zn anode and MnO_2_ cathode can exhibit a higher initial capacity and better cyclic performance than the one with the Zn anode. This study could serve as an essential guideline for inexpensive and easy-to-prepare 3D hosts for high-performance aqueous Zn-ion batteries.

## 2. Results

The fabrication procedure of CF@Zn is shown in Figure 1a. Firstly, the melamine foam was carbonized at 700 °C for 2 h under Ar_2_ to obtain CF, and then Zn was electrochemically deposited into the interior of CF to produce the CF@Zn electrode (Figure 1a). Figure 1b depicts the SEM (scanning electron microscopy) image of melamine foam, and it can be seen that MF exhibits a 3D porous skeleton structure providing a large specific surface area, which is favorable for Zn deposition. However, its conductivity is inferior and cannot be directly used as a zinc anode collector. In this study, a conductive 3D skeleton structure was formed by carbonizing the MF. As shown in Figure 1c, the CF still maintains the intact porous skeleton structure. In addition, energy dispersive spectroscopy (EDS) was carried out for element mapping of fabricated CF. the results of elemental mapping show that carbon, nitrogen, and oxygen elements are uniformly distributed on their surface (Appendix A). This three-dimensional structure provides interconnected channels, facilitating rapid e^−^ transport and excellent conductivity [28,33,35]. The porous 3D carbon foam framework can effectively alleviate the volume expansion of the electrode during repeated charge and discharge cycles, facilitating the integrity of the electrode structure and excellent cycling performance. Hence, a high-performance Zn metal anode could be anticipated by employing this carbon foam. After 10 mAh cm^−2^ amount of zinc deposition, CF@Zn with a relatively smooth surface was obtained (Figure 1d).

Appendix A shows the XRD (X-ray diffraction) patterns of pristine carbon foam and carbon foam after the pre-deposition of 10 mAh cm^−2^ zinc (CF@Zn). The XRD peaks of the CF@Zn sample could be Indexed to the peaks of standard zinc (PDF#04-0831), indicating the successful pre-deposition of Zn ions on the CF surface. Moreover, XPS (X-ray photoelectron spectroscopy) technique was carried out to investigate the surface chemical structures of the CF sample, and the binding energies were referenced to the C1s line at 284.8 eV from adventitious carbon. The results showed that the surface of CF contains carbon, nitrogen, and oxygen elements (Figure 2a), with the highest content of C elements, which is conducive to increase electrical conductivity. Figure 2b depicts the high-resolution C 1s spectra, and the presence of C=N and C-O peaks could be observed. Figure 2c displays the XPS energy spectrum of N 1s with zincophilicity pyrrole nitrogen and pyridine nitrogen peaks. The high-resolution XPS spectrum of O 1s is demonstrated in Figure 2d, where the spectrum consists of two characteristic peaks corresponding to the C=O and C-OH bonds, respectively. According to previous reports in the literature, the presence of pyridine nitrogen, pyrrole nitrogen, C=N [36] and C=O [34] could prove the high zincophilicity on the CF surface. These nitrogen and oxygen-containing functional groups endow the CF with excellent zincophilicity characteristics [14]. The large specific surface area and zincophilicity characteristic of the CF provides substantial plating sites for Zn ions and could effectively regulate uniform Zn deposition [26].

To explore the deposition behavior of zinc ions on CF sample and copper foil at different deposition amounts, scanning electron microscopy was utilized to investigate the electrode morphologies at deposition amounts of 1, 5 and 10 mAh cm^−2^. When plated with a capacity of 1 mAh cm^−2^, the metallic zinc was deposited preferentially on the skeleton, as shown in Figure 3a. As the plating capacity reached 5 mAh cm^−2^, the voids of CF were gradually filled (Figure 3b). When the zinc deposition capacity increased to 10 mAh cm^−2^, the skeleton was fully filled, and the zinc deposition was relatively uniform without the generation of Zn dendrites (Figure 3c). In contrast, at a deposition capacity of only 1 mAh cm^−2^, metallic zinc has spread over the entire Cu foil (Figure 3d). As the deposition capacity continued to increase to 5 mAh cm^−2^ (Figure 3e), holes began to appear on the Cu foil surface. When further increased to 10 mAh cm^−2^, the holes become more apparent (Figure 3f). The different deposition behaviors of zinc on the two collectors indicate that CF can effectively promote the uniform deposition of zinc ions, and there is more space to accommodate zinc ions, thus alleviating the volume expansion during cycling.

To further demonstrate the advantage of the 3D CF for regulating uniform Zn deposition, Zn||Cu foil (or Zn||CF) cells were used to investigate the coulombic efficiency (CE) of Zn anodes. Figure 4a depicts the voltage profiles of the Zn||Cu foil cells at the 1st, 25th and 50th cycles at a current density of 5 mA cm^−2^ and a capacity of 2 mAh cm^−2^. The copper foil electrode exhibited a high overpotential of 88.2 mV. Compared with the copper foil electrode, the CF electrode exhibited a low voltage hysteresis of 60 mV (Figure 4b). In addition, the Zn||CF half-cell is capable of 120 stable cycles, while the Cu foil electrode can only run about 80 cycles (Figure 4c). Compared with the previously reported literature, the average Coulomb efficiency of CF electrodes still has some advantages, especially at the large current density and large areal capacity (Appendix A). In addition, we investigated the surface morphology of CF and Cu foil after cycling a different number of cycles. As shown in Appendix A, after 50 cycles, large Zn dendrites had already formed on the Cu foil electrode. With the cycle number of cycles, more and more Zn flakes could be found on the surface of Cu foil. In striking contrast, the surface of the CF electrode became flat after 50 cycles, and the uniform Zn plating layer, in turn, enabled smooth Zn plating/stripping reactions in the following cycles. When tested at 10 mA cm^−2^ and 10 mAh cm^−2^, the CF electrode exhibited higher cycling stability and longer cycling stability compared to the Cu foil electrode (Figure 4d). The electrochemical performance of the half-cells indicates that the zinc ion plating/stripping process of the carbon foam electrode was more stable, which was closely related to the high specific surface area and porous conductive structure of the carbon foam electrode.

To further investigate the electrochemical behavior of the CF electrode in a symmetric cell, a 10 mAh cm^−2^ capacity of zinc was pre-electrodeposited to the CF and Cu foil electrodes prior to cycling. At a current density of 1 mA cm^−2^ and a capacity of 1 mAh cm^−2^, the symmetric cell with CF electrodes exhibited a low overpotential of about 32.4 mV and was able to cycle stably for nearly 800 h (Figure 5a), which is better than the cycle life of most of the reported zinc anodes with carbon-based hosts or modified by carbon-based materials (Appendix A). On the contrary, with Cu foil electrodes and pure zinc electrodes, the overpotential of the symmetric cells increased after a period of cycling until, finally, an internal short circuit was observed, which may be associated with the generation of zinc dendrites. Furthermore, the cycling stability of the symmetric cell with CF@Zn electrode was tested at a higher current density. Impressively, the symmetric cell with CF@Zn electrodes still exhibited stable cycling (650 h) and low overpotential (45.6 mV) at 2 mA cm^−2^. By contrast, the symmetric cells with Cu foil electrodes and the pure Zn electrodes exhibited significant voltage oscillation and short cycle life (Figure 5b). When increased to 4 mA cm^−2^ and 2 mAh cm^−2^, the symmetric cell with CF electrodes could cycle stably for 500 h with a low voltage polarization (54.1 mV). In contrast, the cell with Cu foil electrodes exhibited a larger overpotential (57.1 mV) and a large voltage drop at 100 h upon cycling, indicating an internal short circuit, followed by severe voltage fluctuations with continued cycling, indicating the generation of substantial zinc dendrites inside the Cu foil electrode. The cell with pure zinc electrodes exhibited a much higher overpotential (76.3 mV) (Figure 5c). Therefore, the above symmetric cell results showed that the CF electrode was able to effectively inhibit the generation of Zn dendrites, ensuring that the cell could be cycled stably for a longer lifetime.

The advantage of the carbon foam skeleton toward stable Zn metal anodes was also demonstrated in full cells by coupling with the MnO_2_ cathode. The X-ray diffraction pattern of the as-prepared cathode material is depicted in Appendix A, which could be indexed to α-MnO_2_. In addition, the scanning electron microscopy image of the as-prepared α-MnO_2_ (Appendix A) depicts the morphology of nanorods. In order to explore the influence of different porosity of CF@Zn on electrochemical performance, we pre-deposited CF with different amounts of Zn (2 mAh cm^−2^, 5 mAh cm^−2^ and 10 mAh cm^−2^) and conducted full-cell electrochemical performance tests of these CF@Zn electrodes. As shown in Appendix A, when the pre-deposition amount of Zn was 2 mAh cm^−2^, the cell with this CF@Zn electrode could not run; when the Zn deposition amount was 5 mAh cm^−2^, the capacity of the cell was relatively low and decreased rapidly; when the deposition amount was 10 mAh cm^−2^, the battery capacity was much higher, and the cycling performance was better than the other two cells, so the deposition amount of 10 mAh cm^−2^ was used in the subsequent electrochemical tests. Figure 6a shows the cyclic voltammetry curves of Zn||α-MnO_2_ and CF@Zn||α-MnO_2_ full cells. Both exhibit essentially similar redox peak positions, indicating that the CF electrode does not change the reaction mechanism of α-MnO_2_. Moreover, the CF@Zn||α-MnO_2_ cell exhibited a higher peak current density and area, indicating that the CF@Zn electrode could provide higher capacity and electrochemical reactivity. The rate performance of Zn||α-MnO_2_ and CF@Zn||α-MnO_2_ full cells were subsequently tested in the current density of 0.5 to 5.0 A g^−1^. As shown in Figure 6b, the average discharge capacities of CF@Zn||α-MnO_2_ full cells were 111.9, 119, 83.9, and 43.5 mAh g^−1^ at 0.5, 1, 2, and 5 A g^−1^. All were higher than those of Zn||α-MnO_2_ full cells. When the current density returned to 0.5 A g^−1^, the CF@Zn||α-MnO_2_ full cell (183.2 mAh g^−1^) remained much higher than the average discharge capacity of the Zn||α-MnO_2_ full cell (128.9 mAh g^−1^). The charge and discharge curves of CF@Zn||α-MnO_2_ and Zn||α-MnO_2_ full cells at different current densities are shown in Figure 6c and Figure 6d, respectively. Compared to the bare Zn||α-MnO_2_ full battery, the CF@Zn||α-MnO_2_ full cell has a lower charging voltage plateau and a higher discharge voltage plateau, indicating that the rate performance of the bare Zn anode is effectively improved. Figure 6e shows the Electrochemical Impedance Spectroscopy (EIS) spectra of the Zn||α-MnO_2_ and CF@Zn||α-MnO_2_ cells. The CF@Zn negative electrode exhibited lower charge transfer resistance (Rct) and zinc ion diffusion resistance, which indicated that the CF electrode had good interfacial wettability with the electrolyte. By fitting the EIS with the equivalent circuit (inset in Figure 6e and Appendix A), the fitting curve matched well with the raw data, indicating the suitability of the employed equivalent circuit. The calculated impendence parameters are shown in Appendix A. The dramatic growth of the Rs and Rct of bare Zn||α-MnO_2_ cell suggest a deteriorated electrode/electrolyte interface during the Zn stripping/plating cycling process. In comparison, for CF@Zn||α-MnO_2_ cells, the Rs and Rct exhibited much slower growth, indicating that the CF could inhibit the formation of Zn dendrites and provide space for volume expansion during repeated charge-discharge cycles. Figure 6f shows the cycling performance of bare Zn||α-MnO_2_ and CF@Zn||α-MnO_2_ at 1 A g^−1^. CF@Zn||α-MnO_2_ exhibited a higher initial capacity (123.6 mAh g^−1^) and still had 66.1 mAh g^−1^ after 180 cycles.

After cycling, the cells were disassembled to check the state of the Zn anodes. As shown in Appendix A, the bare Zn became tattered due to the formation of massive by-products accompanied by dramatic Zn-mass redistribution. On the other hand, the CF@Zn electrode remains quite an integral structure with a relatively uniform surface morphology (Appendix A). Apart from that, the cross-sectional view SEM images of the CF@Zn before and after cycling are shown in Appendix A. The thickness of the CF@Zn electrode before cycling is around 125 µm. After 100 cycles in a full cell with the α-MnO_2_ cathode at 1 A g^−1^, the thickness of the CF@Zn electrode increased a little bit to about 129 µm, which indicates that the abundant zincophilicity functional groups and conductive three-dimensional network of this carbon foam can effectively regulate Zn deposition and alleviate the Zn anode’s volume expansion during cycling. The CF@Zn electrode exhibited a superior full-cell electrochemical performance, which can be attributed to its ability to effectively suppress zinc dendritic growth, thereby improving the cyclic stability of aqueous Zn-ion batteries.

## 3. Materials and Methods

### 3.1. Materials

All reagents were of analytical grade and used as received without further purification. The MA sponge was provided by SINOYQX Co., Ltd., Chengdu, China. KMnO_4_ and MnSO_4_ were purchased from Shanghai Aladdin Biochemical Technology Co., Ltd., Shanghai, China. Ethanol (C_2_H_5_OH, 99.7%) was purchased from Sinopharm Chemical. Reagent Co., Ltd. The Zn foil and CR2032 coin cells were from Guangdong Canrd New Energy Technology Co., Ltd., Dongguan, China.

### 3.2. Preparation of the Carbon Foam

The carbon foam was fabricated via the carbonization method as described elsewhere [33]. Carbon foam (CF) was prepared by calcining melamine foam at 700 °C for 2 h under an Ar_2_ atmosphere (The heating rate was 5 °C min^−1^).

### 3.3. Preparation of the α-MnO_2_ Cathode

For the α-MnO_2_ cathode, 6 mmoL KMnO_4_ was dissolved in 60 mL distilled H_2_O, then 20 mmoL concentrated HCl was slowly added under severe agitation. Afterward, transferred the resulting clear purple solution to a Teflon-lined stainless steel autoclave with a volume of 100 mL. The sealed autoclave was heated to 140 °C for 12 h. After cooling to room temperature, the precipitate was collected through centrifugation and cleaned several times with distilled water. After 12 h dried by vacuum oven at 60 °C, the α-MnO_2_ was obtained.

### 3.4. Electrochemical Measurements

The cathode slurry was obtained by mixing the α-MnO_2_ powder, acetylene black carbon, and polyvinylidene difluoride in a 7:2:1 weight ratio in an NMP solution. Then, the obtained slurry was painted onto the stainless steel mesh. After 12 h of vacuum drying at 60 °C, the stainless steel mesh was cut into 12 mm diameter discs. The obtained discs with a loading mass of 1.0–1.5 mg of active material were used as cathode electrodes for AZIBs. CR2032 coin cells assembled with working and counter electrodes separated via a glass fiber separator (Φ18) were used to assess electrochemical properties. CF pre-deposited 10 mAh cm^−2^ of zinc as the anode (CF@Zn). The performance of Zn plating/stripping was evaluated in symmetrical Zn||Zn (CF@Zn||CF@Zn) batteries with a 2 M ZnSO_4_ (100 µL) solution as the electrolyte. Zn||Cu (or Zn||CF) cells were used to evaluate the CE of Zn anodes. The electrochemical performance of full cells (Zn||α-MnO_2_ or CF@Zn|α-MnO_2_) was investigated using a 2 M ZnSO_4_ + 0.1 M MnSO_4_ (100 µL) solution as the electrolyte at 1 A g^−1^ in the 0.8–1.8 V voltage range. All these tests were performed on a LANDCT2001A battery testing system. Electrochemical impedance spectroscopy (EIS) was evaluated on a CHI660E electrochemical workstation in the 0.01 Hz to 100 kHz frequency range. For full cells, cyclic voltammetry (CV) curves were tested by an electrochemical workstation system (CHI-660E type) with a scan rate of 0.1 mV s^−1^.

### 3.5. Materials Characterizations

X-ray diffraction (XRD) (Bruker D8 ADVANCE, Cu Kα source, a scan speed of 5° min^−1^) was used to determine the phase composition and crystal structure of the prepared samples. Scanning electron microscopy (SEM) was used to evaluate the morphology, energy dispersive spectroscopy (EDS), and element mapping of produced materials (JSM-7800F, JEOL, Akishima, Japan). The chemical structures of the samples were analyzed by X-ray photoelectron spectroscopy (XPS), which was carried out on a Thermo K-Alpha XPS spectrometer (Thermo Fisher Scientific, Waltham, MA, USA) with a monochromatic Al Ka X-ray source (1486.6 eV). The binding energies were referenced to the C1s line at 284.8 eV from adventitious carbon.

## 4. Conclusions

In summary, we have successfully demonstrated the effectiveness of an MF-derived carbon foam collector in the construction of a dendrite-free zinc metal anode. The large specific surface area of carbon foam reduces the local current density of the electrode and avoids the tip effect. Furthermore, the carbon foam surface containing a large number of active sites could promote uniform zinc deposition and inhibit Zn dendritic growth. Symmetric cells assembled with CF@Zn electrodes have superior electrochemical performance than pure Zn electrodes. At 1 mA cm^−2^ and 1 mAh cm^−2^, the CF@Zn symmetric cell depicted a lower overpotential (32.4 mV) and better cycling performance (750 h) than the Zn symmetric cell. After being assembled with α-MnO_2_ into a full cell, the CF@Zn anode exhibited superior rate performance and longer cycling performance than the pure Zn anode. Our further study will be focused on compositing other zincophilic materials (e.g., Ag, ZIF-8 and graphene) with CF to further increase the CF’s electrochemical performance. The development of the CF@Zn anode would bring a new strategy for the realization of high-performance aqueous zinc-ion batteries.

## Figures and Tables

**Figure 1 molecules-28-01742-f001:**
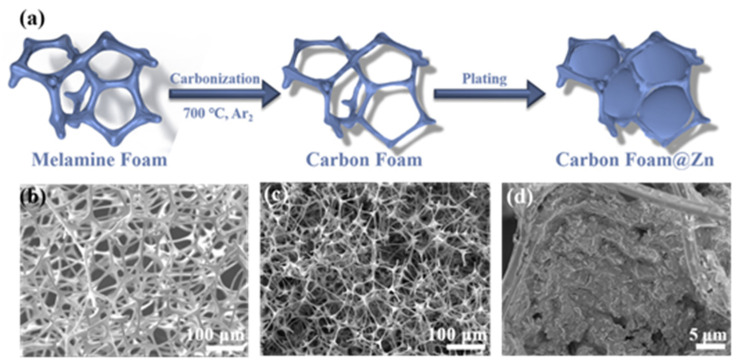
(**a**) Schematic diagram of carbon foam preparation. Scanning electron microscopy image of (**b**) melamine foam; (**c**) Carbon foam (CF); (**d**) Carbon foam@Zn (CF@Zn).

**Figure 2 molecules-28-01742-f002:**
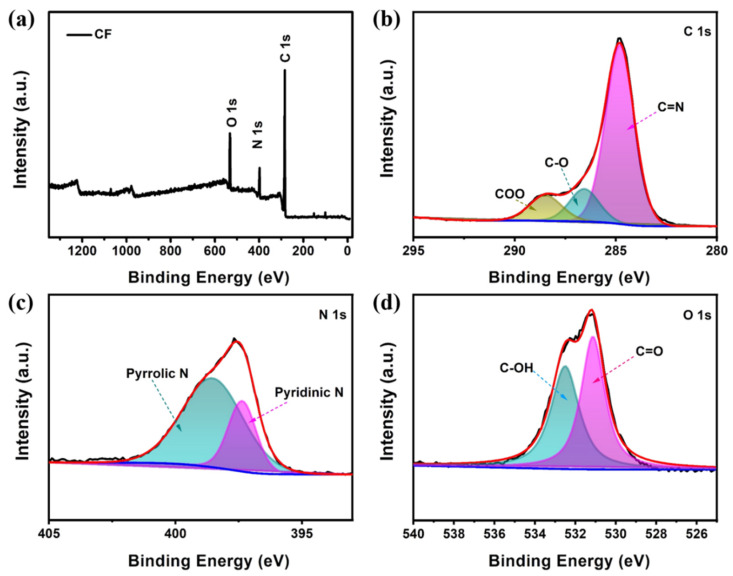
The X-ray photoelectron of carbon foam: (**a**) Survey; (**b**) C 1s; (**c**) N 1s; (**d**) O 1s.

**Figure 3 molecules-28-01742-f003:**
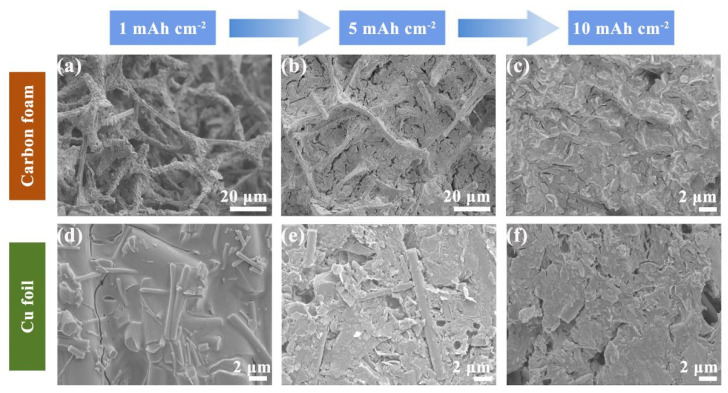
Scanning electron microscopy images of CF and Cu foil electrodes at the deposition current density of 1 mA·cm^−2^ and different deposition capacities: (**a**,**d**) deposition capacity of 1 mAh·cm^−2^; (**b**,**e**) deposition capacity of 5 mAh·cm^−2^; (**c**,**f**) deposition capacity of 10 mAh·cm^−2^.

**Figure 4 molecules-28-01742-f004:**
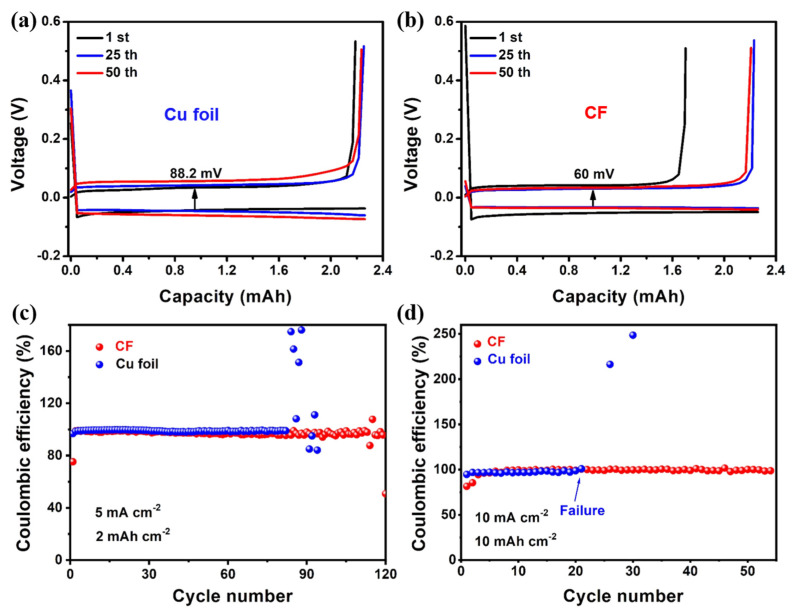
Charge-discharge curves with a current density of 5 mA cm^−2^ and deposition capacity of 2 mAh cm^−2^ for different cycles: (**a**) Cu foil; (**b**) the CF; coulombic efficiency plots for Zn||Cu and Zn||CF batteries at (**c**) 5 mA cm^−2^ with 2 mAh cm^−2^ and (**d**) 10 mA cm^−2^ with 10 mAh cm^−2^.

**Figure 5 molecules-28-01742-f005:**
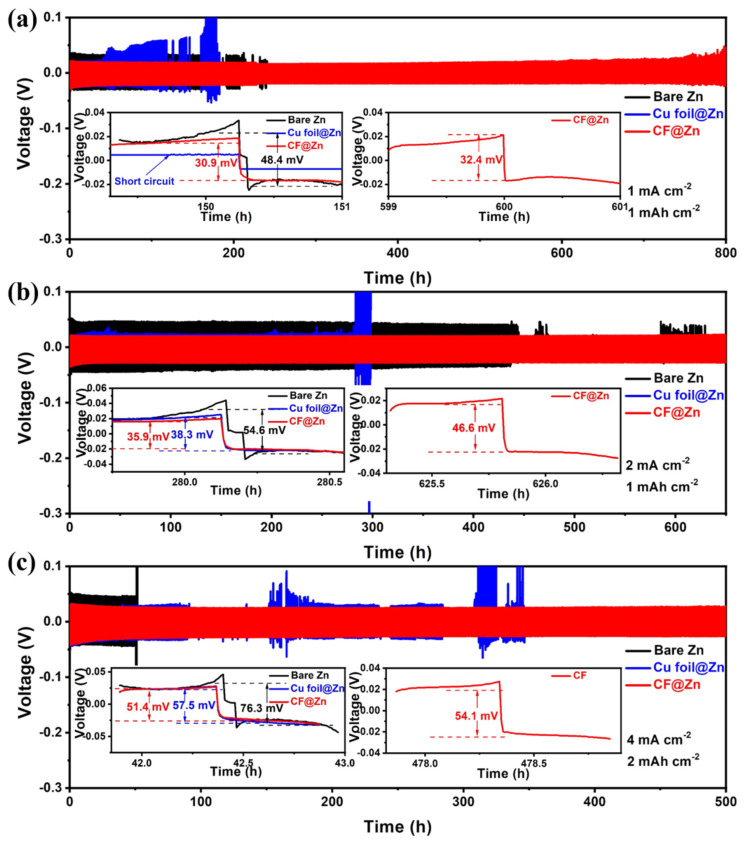
Cyclic performance of symmetric cells with different current densities and deposition capacities: (**a**) 1 mA cm^−2^, 1 mAh cm^−2^; (**b**) 2 mA cm^−2^, 1 mAh cm^−2^; (**c**) 4 mA cm^−2^, 2 mAh cm^−2^ (The insets are the time-voltage curves of the three electrodes at different periods).

**Figure 6 molecules-28-01742-f006:**
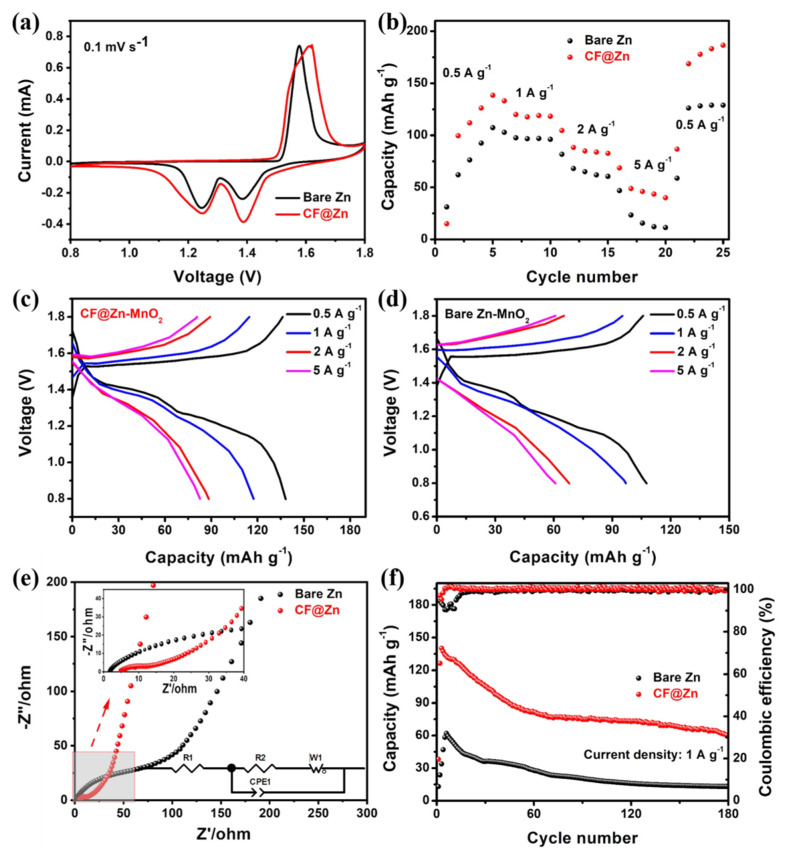
Electrochemical performances of CF@Zn||α-MnO_2_ and Zn||α-MnO_2_ full cells: (**a**) Cyclic voltammetry; (**b**) Rate capability; Charge and discharge curves of (**c**) CF@Zn||α-MnO_2_ and (**d**) Zn||α-MnO_2_ cells. (**e**) Electrochemical impedance spectroscopy of bare Zn and CF@Zn cells before cycling (inset is the equivalent circuit); (**f**) Cycling performance at a current density of 1 A g^−1^.

## Data Availability

The data are contained within this article.

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
