# Peer review of "Melamine Foam-Derived Carbon Scaffold for Dendrite-Free and Stable Zinc Metal Anode"

_molecules, 2023, doi:10.3390/molecules28041742_

Round 1

Reviewer 1 Report

This manuscript reports the preparation of the conductive 3D melamine foam-derived carbon scaffold (CF) and the deposition of Zn in CF, which was utilized as an anode for aqueous Zn-ion batteries. Due to the abundant zincophilicity functional groups and conductive 3D network of the as-prepared CF, the CF@Zn anode could regulate Zn deposition and alleviate volume expansion during cycling, leading to better electrochemical performance compared to bare Zn. This manuscript is interesting, so the reviewer recommends it to publish in Molecules after addressing following points:

1.      It is recommended to provide the XRD pattern of pure CF, from which it can be inferred that the heat-treated CF is carbonized.

2.      Please also provide the reason why 0.1 M MnSO4 was added into the electrolyte.

3.      It is suggested to provide the specific surface area of CF before and after depositing Zn. Based on this, it is essential to discuss the effect of different porosities on the electrochemical performances.

4.      Can the authors provide the SEM images of the electrodes after cycling in order to analyze the difference in Zn deposition on the surface of the electrodes?

5.      There are some typos and grammar issues. For instance,

-          Line 74 : “electrochemical” should be “electrochemically”

Please check the whole manuscript carefully again.

Author Response

Reviewer:This manuscript reports the preparation of the conductive 3D melamine foam-derived carbon scaffold (CF) and the deposition of Zn in CF, which was utilized as an anode for aqueous Zn-ion batteries. Due to the abundant zincophilicity functional groups and conductive 3D network of the as-prepared CF, the CF@Zn anode could regulate Zn deposition and alleviate volume expansion during cycling, leading to better electrochemical performance compared to bare Zn. This manuscript is interesting, so the reviewer recommends it to publish in Molecules after addressing following points:

OUR ANSWER: We thank the reviewer for his/her positive assessment, in particular, and his/her recommending our article for publication in Molecules and we have carefully revised our manuscript according to the valuable comments and suggestions.

Reviewer: It is recommended to provide the XRD pattern of pure CF, from which it can be inferred that the heat-treated CF is carbonized. 

OUR ANSWER: We are thankful to the reviewer for his/her valuable  suggestion. Following the reviewer’s advice, we have added the XRD pattern of pure CF into Fig.S2, shown as follows:

Figure S2. XRD patterns of pure CF and CF@Zn samples.

Reviewer: Please also provide the reason why 0.1 M MnSO4 was added into the electrolyte.

OUR ANSWER: We acknowledge the reviewer. The reason for adding 0.1 M MnSO4 to the electrolyte is to inhibit the dissolution of MnO2 during cycling, thus extending the battery life, which is also widely used in many other related works, such as Journal of Energy Chemistry 55 (2021) 549–556, Energy Storage Materials 27 (2020) 205–211 and Adv. Funct. Mater. 31 (2020) 2001867.

Reviewer: It is suggested to provide the specific surface area of CF before and after depositing Zn. Based on this, it is essential to discuss the effect of different porosities on the electrochemical performances.  

OUR ANSWER: We thank a lot for the reviewer’s valuable comment and suggestion. Because of the Covid-19 and Chinese Spring Festival Vacation, the BET facility is not available currently in and ourside our campus. Unfortunately, we could not carry out the BET tests for CF before and after depositing Zn recently. Following the reviewer’s advice, we have added the full-cell electrochemical performances of pre-deposited CF with different amounts of Zn (2 mAh cm−2, 5 mAh cm−2 and 10 mAh cm−2) and the corresponding discussion in the revised manuscript, which now reads:

“In order to explore the influence of different porosity of CF@Zn on electrochemical performance, we pre-deposited CF with different amounts of Zn (2 mAh cm−2, 5 mAh cm−2 and 10 mAh cm−2), and conducted full-cell electrochemical performance tests of these CF@Zn electrodes. As shown in Figure S5, when the pre-deposition amount of Zn was 2 mAh cm−2, the cell with this CF@Zn electrode could not run; when the Zn deposition amount was 5 mAh cm−2, the capacity of the cell was relatively low and decreased rapidly; when the deposition amount was 10 mAh cm−2, the battery capacity was much higher and the cycling performance was better than the other two cells, so the deposition amount of 10 mAh cm−2 was used in the subsequent electrochemical tests.”Please see page 9.

Figure S5. Cyclic performance of pre-deposited CF with different amounts of Zn (2 mAh cm−2, 5 mAh cm−2 and 10 mAh cm−2) in full cells.

Reviewer: Can the authors provide the SEM images of the electrodes after cycling in order to analyze the difference in Zn deposition on the surface of the electrodes? 

OUR ANSWER: We are thankful to the reviewer for his/her valuable sugguestion. Following the reviewer’s advice, we have added the surface morphology of the CF@Zn and bare Zn anodes after 100 cycles at 1 A g−1 in Fig. S7 in the supporting information.

We have also added the following sentences in the revised manuscript, which now reads:

After cycling, the cells were disassembled to check the state of Zn anodes. As shown in Fig. S7b, the bare Zn became tattered due to the formation of massive by-products accompanied by dramatic Zn-mass redistribution. On the other hand, the CF@Zn electrode remains a quite integral structure with a relatively uniform surface morphology (Fig. S7a). Please see page 10.

Figure S7. SEM images of a) CF@Zn and b) bare Zn anodes in full cells with α-MnO2 cathode after 100 cycles at 1 A g−1.

Reviewer: There are some typos and grammar issues. For instance,-  Line 74 : “electrochemical” should be “electrochemically”Please check the whole manuscript carefully again.

OUR ANSWER: We acknowledge the reviewer for his/her constructive suggestions.

Following the reviewer’s advice, the sentence “then Zn was electrochemical deposited into the interior of CF to produce the CF@Zn electrode” has been revised to “then Zn was electrochemically deposited into the interior of CF to produce the CF@Zn electrode” in the manuscript. Please see page 3.

We have also went through similar issues throughout the manuscript and revised them when deemed necessary. As examples,

The sentence “The large specific surface area and zincthiophilic characteristic of the CF provides substantial plating sites for Zn ions, and could effectively regulate uniform Zn deposition” has been revised to “The large specific surface area and zincophilicity characteristic of the CF provides substantial plating sites for Zn ions, and could effectively regulate uniform Zn deposition” Please see page 4.

Reviewer 2 Report

This manuscript reported a melamine foam-derived carbon scaffold for zinc anode. It is a strategy to regulate Zn deposition and alleviate the Zn anode’s volume expansion during cycling by using the abundant zincophilicity functional groups and conductive three-dimensional network. However, more experimental results need to be provided to support the authors’ claim and there are some issues need to be dealt with:

1.      The introduction could be more precise. In the discussion of the efforts that have been devoted to resolve the problems of Zn anodes, it is recommended to add and discuss some supplement to provide a comprehensively understanding of the existing strategies, such as the studies of carbon-based material coated Zn anodes, including Adv. Mater. 2020, 32, 2001755; J. Colloid Interface Sci. 2021, 594, 389; Energy Material Advances. 2022. 2022, 9809626, doi.org/10.34133/2022/9809626.

2.      The author mentioned the price of 3D graphene-fiber host is high and the preparation process is complicated, while the melamine foam that used in this work is inexpensive. Please provide detailed analysis of the cost.

3.      Though melamine foam derived CF has been rarely reported, it is not a strong enough motivation to conduct this work. Please provide more advantages of melamine foam derived CF compared with the existing strategies.

4.      According to Figure 1(a), it seems Zn deposit in the voids of the carbon foam. Please confirm the deposition mechanism by EDS.

5.      There are typos in the manuscript. For example, ‘zincthiophilic’ in page 3 should be zincophilicity.

6.      Generally, the sample surface gets charged while doing XPS, and the data is possible to shift either towards the lower or higher binding energy, so it is necessary to do calibration before XPS analysis. There is no calibration information in the experimental section, please explain. 

7.      As displayed in Figure 4, the Zn//Cu cell fails at about 80th cycle, it is recommended to provide the morphologies of the Cu and CF electrodes after multiple cycles, to investigate the failure behavior of Cu and prove the merits of CF electrode. Please also provide the CE values of Zn//CF and Zn//Cu cells, and compare them with values obtained by the existing strategies.

8.      Figure 5 shows the cyclic performance of symmetric cells with the as-made electrodes, please compare the performance with the existing strategies, especially the Zn anode with carbon-based coating materials and the utilization of other carbon-based scaffolds.

9.      In the abstract, the authors mentioned the expansion of Zn during cycling could be alleviated by using this CF scaffold, please show the experimental evidence.

10.  More EIS results are required. There are only EIS of pristine cells in Figure 6, how about the cells after cycling? It is also suggested to fit the EIS data and add the equivalent circuit.

Author Response

Reviewer: This manuscript reported a melamine foam-derived carbon scaffold for zinc anode. It is a strategy to regulate Zn deposition and alleviate the Zn anode’s volume expansion during cycling by using the abundant zincophilicity functional groups and conductive three-dimensional network. However, more experimental results need to be provided to support the authors’ claim and there are some issues need to be dealt with:

OUR ANSWER: We thank the reviewer for his/her assessment. We have carefully revised the manuscript according to his/her valuable comments and suggestions.

Reviewer:  The introduction could be more precise. In the discussion of the efforts that have been devoted to resolve the problems of Zn anodes, it is recommended to add and discuss some supplement to provide a comprehensively understanding of the existing strategies, such as the studies of carbon-based material coated Zn anodes, including Adv. Mater. 2020, 32, 2001755; J. Colloid Interface Sci. 2021, 594, 389; Energy Material Advances. 2022. 2022, 9809626, doi.org/10.34133/2022/9809626.

OUR ANSWER: We are thankful to the reviewer for his/her comments. Following the reviewer’s advice, we have added the the related work (Adv. Mater. 2020, 32, 2001755; J. Colloid Interface Sci. 2021, 594, 389; Energy Material Advances. 2022. 2022, 9809626) in the revised manuscript, which now reads:

To tackle this problem, some carbon-based materials have been coated or composted with Zn anodes, such as hydrogen-substituted graphdiyne [14], zinc microspheres/carbon nanotubes/nanocellulose composite film [15], and carbon-coated NaTi2(PO4)3 [16], successfully regulating uniform Zn deposition.” Please see Page 2.

Furthermore, we also add some supplement of the existing strategies, “novel separator design” in Page 2.

Reviewer:The author mentioned the price of 3D graphene-fiber host is high and the preparation process is complicated, while the melamine foam that used in this work is inexpensive. Please provide detailed analysis of the cost. 

OUR ANSWER: We are thankful to the reviewer for his/her comments. We do apologize that we made a mistake here. The example of 3D host design for Zn anode has been revised. Following the reviewer’s advice, we introduced the related price in the revised manuscript, which now reads: 

Recently, Lin and his co-workers[31] fabricated freestanding 3D Zn-graphene anode to mitigate Zn dendrite growth. Unfortunately, the high price (~ ¥ 132 g−1, price from Suzhou Tanfeng Graphene Technology Co., Ltd) and complicated preparation process of this 3D Zn-graphene anode limit its large-scale application.” Please see page 2.

Melamine foam, a commercially available material, has been widely used in the chemical industry due to its low cost  (~ ¥ 2.6 g−1, price from SINOYQX Co. Ltd., Sichuan, China), low density, high porosity, and open structures [29].” Please see page 2.

Reviewer: Though melamine foam derived CF has been rarely reported, it is not a strong enough motivation to conduct this work. Please provide more advantages of melamine foam derived CF compared with the existing strategies.

OUR ANSWER: We are thankful to the reviewer for his/her comments. Following the reviewer’s advice, we have added the advantages of melamine foam (MF) and MF derived CF in the revised manuscript, which now reads:

“Melamine foam, a commercially available material, has been widely used in the chemical industry due to its low cost (~ ¥ 2.6 g−1, price from SINOYQX Co. Ltd., Sichuan, China), low density, high porosity, and open structures[32]. The melamine foam derived CF has rich N-containing functional groups, such as pyridine nitrogen, pyrrole nitrogen and C=O[33], which have been proven to be highly zincophilic [34]. And these N-containing functional groups on CF would effectively control uniform Zn deposition.” Please see page 2-3.

Reviewer: According to Figure 1(a), it seems Zn deposit in the voids of the carbon foam. Please confirm the deposition mechanism by EDS.

OUR ANSWER: We are thankful to the reviewer for his/her comments. Unfortunately, the EDS on our school’s SEM was down several weeks ago, due to the  Covid-19 and Chinese Spring Festival Vacation, the EDS could not be fixed recently. So the EDS for the CF@Zn sample could not be supplemented in a short time.

Reviewer: There are typos in the manuscript. For example, ‘zincthiophilic’ in page 3 should be zincophilicity.

OUR ANSWER: We are thankful to the reviewer for his/her comments.

Following the reviewer’s advice, the sentence “The large specific surface area and zincthiophilic characteristic of the CF provides substantial plating sites for Zn ions, and could effectively regulate uniform Zn deposition” has been revised to “The large specific surface area and zincophilicity characteristic of the CF provides substantial plating sites for Zn ions, and could effectively regulate uniform Zn deposition” Please see page 4.

We have also went through similar issues throughout the manuscript and revised them when deemed necessary. As examples,

The sentence “then Zn was electrochemical deposited into the interior of CF to produce the CF@Zn electrode” has been revised to “then Zn was electrochemically deposited into the interior of CF to produce the CF@Zn electrode” in the manuscript. Please see page 3.

Reviewer: Generally, the sample surface gets charged while doing XPS, and the data is possible to shift either towards the lower or higher binding energy, so it is necessary to do calibration before XPS analysis. There is no calibration information in the experimental section, please explain.

OUR ANSWER: We thank a lot for the comment and suggestion. Following the reviewer’s advice, we have added the calibration information in the experimental section, which now reads:

The chemical structures of the samples were analyzed by X-ray photoelectron spectroscopy (XPS), which was carried out on a Thermo K-Alpha XPS spectrometer (Thermo Fisher Scientific) with a monochromatic Al Ka X-ray source (1486.6 eV). The binding energies were referenced to the C1s line at 284.8 eV from adventitious carbon.” Please see page 12.

Reviewer:  As displayed in Figure 4, the Zn//Cu cell fails at about 80th cycle, it is recommended to provide the morphologies of the Cu and CF electrodes after multiple cycles, to investigate the failure behavior of Cu and prove the merits of CF electrode. Please also provide the CE values of Zn//CF and Zn//Cu cells, and compare them with values obtained by the existing strategies.

OUR ANSWER: We are thankful to the reviewer for his/her constructive suggestions. Following the reviewer’s advice, we have added the surface morphology of the Cu and CF electrodes after multiple cycles in Fig. S3 in the supporting information.

Figure S3. Scanning electron microscopy images of (a,c,e) CF and (b,d,f) Cu foil electrodes at the deposition current density of 5 mA·cm−2 and deposition capacities of 2 mAh·cm−2 with different cycles: (a) and (b) 20 cycles; (c) and (d) 50 cycles; (e) and (f) 80 cycles.

We have added the following sentences in the revised manuscript, which now reads:

“In addition, we investigated the surface morphology of CF and Cu foil after cycling different number of cycles. As shown in Fig. S3, after 50 cycles, large Zn dendrites had already formed on the Cu foil electrode. With the cycle number of cycles, more and more Zn flakes could be found on the surface of Cu foil. On a striking contrast, the surface of CF electrode became flat after 50 cycling, and the uniform Zn plating layer in turn enables smooth Zn plating/stripping reactions in the following cycles.” Please see page 6.

We also provide the CE values of Zn//CF and Zn//Cu cells, and compare them with values obtained by the existing strategies in the revised manuscript, which now reads:

“Compared with the previously reported literature, the average Coulomb efficiency of CF electrodes still has some advantages, expecially at the large current density and large areal capacity (Table S2).” Please see page 6.

Table S2: Different host materials for zinc metal anodes and their corresponding average CE values.

Skeletons

Current

density

(mA cm−2)

Areal

Capacity (mAh cm−2)

Average Coulombic Efficiency

Reference

Ti3C2Tx MXene

1

1

94.13%

[1]

Graphite felt

1

1

96.5%

[2]

Cu foil

CF

5

2

83.1%

This work

This work

5

2

93.25%

Reviewer: Figure 5 shows the cyclic performance of symmetric cells with the as-made electrodes, please compare the performance with the existing strategies, especially the Zn anode with carbon-based coating materials and the utilization of other carbon-based scaffolds.

OUR ANSWER: We thank a lot for the comment and suggestion. Following the reviewer’s advice, We added a comparison of the symmetric cell performance with the reported literature in the revised manuscript. 

At 1 mA cm−2 and 1 mAh cm−2, the symmetric cell with CF electrodes exhibited a low overpotential of about 32.4 mV and was able to cycle stably for nearly 800 h (Fig. 5a), which is better than the cycle life of most of the reported zinc anodes with carbon-based hosts or modified by carbon-based materials (Table S1, Supporting Information).” Please see page 7.

Table S1: A survey of zinc anodes with carbon-based hosts or modified by carbon-based materials and corresponding electrochemical properties.

Zinc anodes

Current

density

(mA cm−2)

Areal

Capacity (mAh cm−2)

Voltage hysteresis

(V)

Worked

time (h)

Reference

rGO@Zn

2

2

0.1

200

[3]

graphite-coated Zn anode

0.1

0.1

0.028

200

[4]

Zn/C3N4

2

2

/

500

[5]

ZF@CB-NFC

0.5

0.5

0.160

400

[6]

CF@Zn

1

1

0.032

800

This work

2

1

0.047

700

4

2

0.054

500

Reviewer: In the abstract, the authors mentioned the expansion of Zn during cycling could be alleviated by using this CF scaffold, please show the experimental evidence.

OUR ANSWER: We thank a lot for the comment and suggestion. Following the reviewer’s advice, We added cross-sectional view SEM images of CF@Zn electrode before and after the cycle in the revised manuscript. 

“Apart from that, the cross-sectional view SEM images of the CF@Zn before and after cycling are shown in Fig. S8. The thickness of the CF@Zn electrode before cycling is around 125 µm. After 100 cycles in full cells with α-MnO2 cathode at 1 A g−1, the thickness of CF@Zn electrode increased a little to about 129 µm, which indicates that the CF electrode can effectively mitigate the volume expansion.” Please see page 10.

Figure S8. Cross-sectional view SEM images of  CF@Zn anodes a) before and b) after 100 cycles in full cells with α-MnO2 cathode at 1 A g−1.

Reviewer: More EIS results are required. There are only EIS of pristine cells in Figure 6, how about the cells after cycling? It is also suggested to fit the EIS data and add the equivalent circuit.

OUR ANSWER: We are thankful to the reviewer for his/her comments. Following the reviewer’s advice, we have supplemented the EIS data after cycling and fit the EIS data as well as added the equivalent circuit.

We have added the following sentences in the revised manuscript, which now reads:

“By fitting the EIS with the equivalent circuit (inset in Figure 6e and Figure S6), the fitting curve matched well with the raw data, indicating the suitability of the employed equivalent circuit. The calculated impendence parameters were shown in Table S3 and Table S4. The dramatical growth of the Rs and Rct of bare Zn||α-MnO2 cell suggest a deteriorated electrode/electrolyte interface  during the Zn stripping/plating cycling process. In comparison, for CF@Zn||α-MnO2 cell, the Rs and Rct exhibited much slower growth, indicating that the CF could inhibit the formation of Zn dendrites and provide space for volume expansion during cycling.” Please see page 9-10.

Figure 6. Electrochemical performances of CF@Zn||α-MnO2 and Zn||α-MnO2 full cells: (a) Cyclic voltammetry; (b) Rate capability; Charge and discharge curves of (c) CF@Zn||α-MnO2 and (d) Zn||α-MnO2 cells. (e) Electrochemical impedance spectroscopy of bare Zn and CF@Zn cells before cycling (inset is the equivalent circuit); (f) Cycling performance at current density of 1 A g−1.

Figure S6. Nyquist plots of bare Zn and CF@Zn full cells at 1 A g−1after 100 cycles   (inset is the equivalent circuit).

Table S3: The impedance parameters for full cells with α-MnO2 cathode before cycling.

Anodes

Rs (Ω)

Rct (Ω)

CF@Zn

4.507

5.671

Bare Zn

5.145

102

Table S4: The impedance parameters for full cells with α-MnO2 cathode at 1 A g−1after 100 cycles.

Anodes

Rs (Ω)

Rct (Ω)

CF@Zn

4.61

15.86

Bare Zn

5.39

225.9

Reviewer 3 Report

In this manuscript, a carbon foam (CF) was fabricated via melamine-foam carbonization as a scaffold for dendrite-free and stable Zn anode. The conductive three-dimensional network of this carbon foam can effectively regulate Zn deposition and alleviate the Zn anode’s volume expansion during cycling. The full battery coupling CF@Zn anode with MnO2 cathode can exhibit a higher initial capacity and better cyclic performance than the one with the bare Zn anode. This work brings a new idea for the design of three-dimensional (3D) current collectors for stable zinc metal anode towards high-performance AZIBs. Therefore, this paper has certain advantages for this field research and has value to be published in molecules after modification. The comments and suggestions about this work are described as follows:

1.     Many of the figures in this manuscript are not clear enough, please upload high-definition figures.

2.     Please explain the phenomenon of the large increase in the rate capability test in figure 6b at 0.5 A g-1 and the large decrease in the capacity in figure 6f.

3.     The illustrations in figure 5 are so crowded that the coordinates is not clear. Please adjust it.

4.     The figures in the manuscript lacks the coordinate units, and there are many syntax errors, please check carefully and correct.

5.     Some interesting papers on Zn anode protection can be discussed in this manuscript, such as Nano-Micro Lett. 2022, 14:218; Natl. Sci. Rev., 2022, 9, nwab177; Chinese J. Inorg. Chem., 2022, 38, 1451; etc.

Author Response

Reviewer: In this manuscript, a carbon foam (CF) was fabricated via melamine-foam carbonization as a scaffold for dendrite-free and stable Zn anode. The conductive three-dimensional network of this carbon foam can effectively regulate Zn deposition and alleviate the Zn anode’s volume expansion during cycling. The full battery coupling CF@Zn anode with MnO2 cathode can exhibit a higher initial capacity and better cyclic performance than the one with the bare Zn anode. This work brings a new idea for the design of three-dimensional (3D) current collectors for stable zinc metal anode towards high-performance AZIBs. Therefore, this paper has certain advantages for this field research and has value to be published in molecules after modification. The comments and suggestions about this work are described as follows:

OUR ANSWER: We thank the reviewer for his/her supporting our manuscript for publication in the molecules. We have carefully revised the manuscript according to his/her valuable comments and suggestions.

Reviewer: Many of the figures in this manuscript are not clear enough, please upload high-definition figures. 

OUR ANSWER: We are thankful to the reviewer for his/her comments. Following the reviewer’s advice, we increased the resolution of the figures in the revised manuscript.

Reviewer: Please explain the phenomenon of the large increase in the rate capability test in figure 6b at 0.5 A g-1 and the large decrease in the capacity in figure 6f.

OUR ANSWER: We acknowledge the reviewer. The large increase in the rate capability test in figure 6b at 0.5 A g-1 may be associated with the activation of α-MnO2 in the initial 5 cycles; Compare to the cell with bare Zn anode, the cell with CF@Zn anode exhibited much better cycling stability. With the cycling continues, the capacity of CF@Zn cell decreased, which may be associated with the large porosity of the CF and relatively limited zincophilicity. So, our future work will be focused on further increasing the zincophilicity of CF electrode by surface modification to increase it cycling stability.

Reviewer: The illustrations in figure 5 are so crowded that the coordinates is not clear. Please adjust it.

OUR ANSWER: We acknowledge the reviewer for his/her constructive suggestions.

Following his/her advice, we have adjusted Figure 5, shown as follows:

Figure 5. Cyclic performance of symmetric cells with different current densities and deposition capacities: (a) 1 mA cm−2, 1 mAh cm−2; (b) 2 mA cm−2, 1 mAh cm−2; (c) 4 mA cm−2, 2 mAh cm−2 (The insets are the time-voltage curves of the three electrodes at different periods)

Reviewer: The figures in the manuscript lacks the coordinate units, and there are many syntax errors, please check carefully and correct.

OUR ANSWER: We acknowledge the reviewer for his/her constructive suggestions.

Following the reviewer’s advice, the figures in the manuscript has been added the coordinate units.

The sentence “then Zn was electrochemical deposited into the interior of CF to produce the CF@Zn electrode” has been revised to “then Zn was electrochemically deposited into the interior of CF to produce the CF@Zn electrode” in the manuscript. Please see page 3.

The sentence “The large specific surface area and zincthiophilic characteristic of the CF provides substantial plating sites for Zn ions, and could effectively regulate uniform Zn deposition” has been revised to “The large specific surface area and zincophilicity characteristic of the CF provides substantial plating sites for Zn ions, and could effectively regulate uniform Zn deposition” Please see page 4.

.

Reviewer: Some interesting papers on Zn anode protection can be discussed in this manuscript, such as Nano-Micro Lett. 2022, 14:218; Natl. Sci. Rev., 2022, 9, nwab177; Chinese J. Inorg. Chem., 2022, 38, 1451; etc.

OUR ANSWER: We are thankful to the reviewer for his/her comments. Following the reviewer’s advice, we have added the the related work (Nano-Micro Lett. 2022, 14:218; Natl. Sci. Rev., 2022, 9, nwab177; Chinese J. Inorg. Chem., 2022, 38, 1451) in the revised manuscript, please see Ref.20, Ref.24 and Ref.19, respectively.

Round 2

Reviewer 1 Report

The authors revised the manuscript based on the reviewers' comments. Therefore, it can be published in Molecules in its present form.

Reviewer 2 Report

Comments have been well addressed.